# The Xanthine Derivative KMUP-1 Inhibits Hypoxia-Induced TRPC1 Expression and Store-Operated Ca^2+^ Entry in Pulmonary Arterial Smooth Muscle Cells

**DOI:** 10.3390/ph17040440

**Published:** 2024-03-29

**Authors:** Zen-Kong Dai, Yi-Chen Chen, Su-Ling Hsieh, Jwu-Lai Yeh, Jong-Hau Hsu, Bin-Nan Wu

**Affiliations:** 1Department of Pediatrics, School of Medicine, College of Medicine, Kaohsiung Medical University, Kaohsiung 807, Taiwan; zenkong@kmu.edu.tw (Z.-K.D.); jhh936@yahoo.com.tw (J.-H.H.); 2Division of Pediatric Cardiology and Pulmonology, Department of Pediatrics, Kaohsiung Medical University Hospital, Kaohsiung 807, Taiwan; 3Department of Pharmacology, Graduate Institute of Medicine, College of Medicine, Kaohsiung Medical University, Kaohsiung 807, Taiwan; 770137@mail.kmuh.org.tw (Y.-C.C.); jwulai@kmu.edu.tw (J.-L.Y.); 4Department of Pharmacy, Kaohsiung Medical University Hospital, Kaohsiung 807, Taiwan; soolinchia@yahoo.com.tw; 5Department of Medical Research, Kaohsiung Medical University Hospital, Kaohsiung 807, Taiwan

**Keywords:** canonical transient receptor potential channel 1, hypoxia, KMUP-1, protein kinases, pulmonary artery smooth muscle cells, store-operated calcium entry

## Abstract

Exposure to hypoxia results in the development of pulmonary arterial hypertension (PAH). An increase in the intracellular Ca^2+^ concentration ([Ca^2+^]_i_) in pulmonary artery smooth muscle cells (PASMCs) is a major trigger for pulmonary vasoconstriction and proliferation. This study investigated the mechanism by which KMUP-1, a xanthine derivative with phosphodiesterase inhibitory activity, inhibits hypoxia-induced canonical transient receptor potential channel 1 (TRPC1) protein overexpression and regulates [Ca^2+^]_i_ through store-operated calcium channels (SOCs). Ex vivo PASMCs were cultured from Sprague-Dawley rats in a modular incubator chamber under 1% O_2_/5% CO_2_ for 24 h to elucidate TRPC1 overexpression and observe the Ca^2+^ release and entry. KMUP-1 (1 μM) inhibited hypoxia-induced TRPC family protein encoded for SOC overexpression, particularly TRPC1. KMUP-1 inhibition of TRPC1 protein was restored by the protein kinase G (PKG) inhibitor KT5823 (1 μM) and the protein kinase A (PKA) inhibitor KT5720 (1 μM). KMUP-1 attenuated protein kinase C (PKC) activator phorbol 12-myristate 13-acetate (PMA, 1 μM)-upregulated TRPC1. We suggest that the effects of KMUP-1 on TRPC1 might involve activating the cyclic guanosine monophosphate (cGMP)/PKG and cyclic adenosine monophosphate (cAMP)/PKA pathways and inhibiting the PKC pathway. We also used Fura 2-acetoxymethyl ester (Fura 2-AM, 5 μM) to measure the stored calcium release from the sarcoplasmic reticulum (SR) and calcium entry through SOCs in hypoxic PASMCs under treatment with thapsigargin (1 μM) and nifedipine (5 μM). In hypoxic conditions, store-operated calcium entry (SOCE) activity was enhanced in PASMCs, and KMUP-1 diminished this activity. In conclusion, KMUP-1 inhibited the expression of TRPC1 protein and the activity of SOC-mediated Ca^2+^ entry upon SR Ca^2+^ depletion in hypoxic PASMCs.

## 1. Introduction

Pulmonary arterial hypertension (PAH) is characterized by progressive pulmonary arterial resistance and elevation of pulmonary artery pressure (PAP). The increased pulmonary vascular resistance contributes to right ventricular failure, poor prognosis, and impaired quality of life [1]. The pathogenesis and pathophysiology of PAH remain elusive. They include a rare dyspnea-fatigue syndrome due to a progressive increase in pulmonary vascular resistance (PVR) and eventually right ventricular (RV) failure [2]. The vascular lesions in PAH are characterized by vasocontraction, smooth muscle and endothelial cell proliferation, and thrombus formation [1,2]. The imbalance in vasoactive mediators is critical in developing and progressing the obstructive proliferative pathological changes of the distal pulmonary arteries [3]. When untreated, it leads to heart failure and premature death [4]. The precise mechanism that contributes to the development, exacerbation, and treatment of PAH remains to be investigated.

A definitive diagnosis of PAH necessitates ruling out any underlying cardiac, pulmonary, or thromboembolic factors that may lead to an increase in PAP and then conducting a right heart catheterization showing a mean PAP higher than 20 mmHg, pulmonary capillary wedge pressure (PCWP) lower than or equal to 15 mmHg, and PVR higher than or equal to 3 Wood units [1,5,6]. The sharp increase of PAP caused by hypoxia-induced vascular constriction is one of the most critical pathogeneses of high-altitude pulmonary edema [7,8]. Due to limited research on human beings, relevant animal and cell models are needed to investigate the molecular mechanisms of pulmonary vascular remodeling processes.

In PAH, disruption of the homeostasis of intracellular Ca^2+^ ([Ca^2+^]_i_) can partly describe the dysfunction of pulmonary artery smooth muscle cells (PASMCs). One of the most essential players in modulating Ca^2+^ homeostasis is store-operated Ca^2+^ channels (SOCs) that mediate store-operated Ca^2+^ entry (SOCE) [9]. Generally, people recognize that SOCs regulate vascular smooth muscle (VSM) contraction and cell proliferation in the resistant vasculature [9,10]. Similarly, SOCs have also been demonstrated to modulate the pulmonary vascular tone and PASMC proliferation [11]. Indeed, people are interested in SOCs regulated in PASMCs for severe pulmonary arteriopathies [12]. Until now, research on PASMCs has apparently emphasized the role of PIP2, IP3, and other lipid products in modulating those channels. Surprisingly, only certain pulmonary studies have gone beyond SOC regulation and investigated the role of protein kinase C (PKC), protein kinase A, and protein kinase G. In PASMCs, SOCs are shown to be stimulated by PKC activation and PKA and PKG inhibition [13].

SOCs are calcium-selective cation channels representing a major pathway for calcium signaling throughout the body. The essential function of SOCs in maintaining calcium homeostasis is directly tied to their physical and functional association with the endoplasmic reticulum (ER), which triggers calcium influx in response to ER Ca^2+^ depletion [14]. Upon depletion of Ca^2+^ in the ER/SR, SOCs in the plasma membrane are activated, causing capacitative Ca^2+^ entry (CCE). This exceptional Ca^2+^ influx mechanism supports the maintenance of elevated cytoplasmic Ca^2+^ concentration ([Ca^2+^]_cyt_) and is also crucial for refilling Ca^2+^ storage [15].

Canonical transient receptor potential channel 1 (TRPC1) is a non-selective cation channel permeable to monovalent and divalent cations. TRPC1 has a role in activating SOCE that is not direct but requires the interplay between stromal interaction molecule 1 (STIM1) and Orai1 [16,17,18]. SOCE is an essential mechanism of calcium influx expressed in excitable and non-excitable cells. TRPC1-based SOCE contributes to VSM contraction, proliferation, and migration, a potential target for cardiovascular diseases [14,17]. SOCE is also responsible for the development and progression of PAH [17,19,20]. Transient receptor potential (TRP) channels have gained considerable attention, focusing on the role of TRPC1 and other TRPC channels as candidates for conducting Ca^2+^ influx during SOCE [14]. It is generally agreed that SOCE is one of the critical factors leading to pulmonary vascular smooth muscle functional and structural changes during hypoxia [14,21,22].

PAH remains incurable despite therapeutic advances during recent decades [23]. New therapies and preventative strategies to lower the impact of PAH are significantly needed. Thus, we are searching for and developing new agents that can target PAH. Our previous findings showed that the xanthine derivative KMUP-1 (7-[2-[4-(2-chlorobenzene)piperazinyl]ethyl]-1,3-dimethylxanthine, Figure 1A) is recognized to elevate PKA and PKG and stimulate K^+^ channels, leading to the relaxation of smooth muscles. KMUP-1 decreases cardiac hypertrophy via the NO/cGMP/PKG pathway and prevents PAH via K^+^-channel activation and Rho kinase inhibition. It has also been demonstrated to improve monocrotaline-induced PAH by modulating Ca^2+^ sensitization and K^+^-channel [24]. In rat basilar arteries, KMUP-1 was shown to activate large-conductance Ca^2+^-activated K^+^ (BK_Ca_) channels and inhibit L-type calcium channels (LTCCs). In addition to BK_Ca_ channel activation, KMUP-1 depresses the transient and late sodium current (I_Na_) components in GH_3_ pituitary cells [25]. Furthermore, KMUP-1 has been confirmed to prevent subarachnoid hemorrhage-induced cerebral vasospasm in rats, attributed to its stimulation in K^+^ channels [26]. Emerging evidence displays that the activation of the K^+^-channel opening is considered to be associated with antinociceptive actions [27]. KMUP-1 was proven to restore the peripheral nerve injury-induced BK_Ca_ channel inhibition in dorsal root ganglia [28]. This study aims to investigate how the K^+^ channel activator KMUP-1 modulates protein kinases to protect against hypoxic PASMCs that mimic PAH.

## 2. Results

### 2.1. Effects of KMUP-1 in Normoxic and Hypoxic PASMCs

The expression of the TRPC family is encoded for store-operated calcium channels in hypoxic pulmonary arterial smooth muscle cells. A previous report showed that TRPC1 is an important Ca^2+^-permeable channel that mediates pulmonary vasoconstriction when PASMC intracellular Ca^2+^ stores are depleted [15,21]. Thus, we investigated the effects of KMUP-1 incubated in normoxia and hypoxia mediums for 24 h to observe the expression of TRPC1. TRPC1 protein was markedly increased under hypoxic conditions, but there were no changes with normoxia. KMUP-1 (1, 10, 100 μM) did not influence TRPC1 under normoxia (Figure 1B) but significantly attenuated hypoxia-enhanced TRPC1 (Figure 1C).

### 2.2. KMUP-1 Prevented Hypoxia-Enhanced TRPC1 Expression through cGMP/PKG Activation

As shown in Figure 2, the PKG inhibitor KT5823 increased the expression of TRPC-1 in a hypoxic medium. KMUP-1 attenuated hypoxia-induced TRPC1 protein, and the effect was restored by KT5823. A membrane-permeable analog of cGMP, 8-Br-cGMP, is a PKG activator that also suppresses hypoxia-enhanced TRPC1 protein, whereas KMUP-1 combined with 8-Br-cGMP appears not to affect this response significantly. Thus, we suggest that the effects of KMUP-1 could be due to the activation of the cGMP/PKG pathway to attenuate hypoxia-enhanced TRPC1 protein.

### 2.3. KMUP-1 Prevented Hypoxia-Enhanced TRPC1 Expression through cAMP/PKA Activation

Like PKA5823, PKA inhibitor KT5720 increased the expression of TRPC-1 in the hypoxia state. KMUP-1 lessens hypoxia-induced TRPC1 protein, and it can be restored by KT5720. The PKA activator 8-Br-cAMP weakened hypoxia-enhanced TRPC1 protein expression, while KMUP-1 combined with 8-Br-cAMP did not markedly influence this effect (Figure 3). The data show that KMUP-1 reduction of hypoxia-enhanced TRPC1 protein could be involved in activating the cAMP/PKA pathway.

### 2.4. KMUP-1 Prevented Hypoxia-Enhanced TRPC1 Expression through PKC Inhibition

A PKC inhibitor, chelerythrine, suppressed the expression of TRPC-1 under hypoxic conditions. KMUP-1 reduced hypoxia-induced TRPC1 protein, which was restored by a PKC activator, PMA. A PKC inhibitor, chelerythrine, suppressed hypoxia-enhanced TRPC1 protein, and KMUP-1 combined with chelerythrine had no significant effects on this (Figure 4). The data indicate that KMUP-1-retarded hypoxia-enhanced TRPC1 protein could be involved in inhibiting the PKC pathway.

### 2.5. Enhanced Capacitative Ca^2+^ Entry (CCE) in Hypoxic PASMCs

PASMCs were loaded with 5 µM Fura-2-AM for 30 min in a Ca^2+^-free PBS solution. Next, we added 1 µM SERCA inhibitor thapsigargin (TG) and 5 µM VOCC inhibitor nifedipine to deplete intracellular Ca^2+^ stores and block the L-type Ca^2+^ channels, and then 2.5 mM Ca^2+^ was applied to Fura-2-loaded PASMCs. As Figure 5 shows, hypoxia did not affect the sarcoplasmic reticulum (SR) Ca^2+^ release but significantly increased the capacitative Ca^2+^ entry (CCE) via the SOCs in PASMCs.

### 2.6. KMUP-1 Attenuated SR Ca^2+^ Release and SOCs-Mediated CCE in Hypoxic PASMCs

In the hypoxic Ca^2+^-free PBS solution, Fura-2-loaded PASMCs were incubated with 1 µM TG and 5 µM nifedipine in the presence or absence of KMUP-1 (1, 10, 100 μM). KMUP-1 produced markedly dose-dependent (10, 100 μM) decreases in the SR-mediated Ca^2+^ release except at 1 μM. After that, 2.5 mM Ca^2+^ was added to induce the CCE via the SOCs. KMUP-1 (1, 10, 100 μM) significantly attenuated the CCE in a dose-dependent manner (Figure 6).

### 2.7. PKA/PKG/PKC Involvement in KMUP-1-Attenuated SOCs-Mediated CCE in Hypoxic PASMCs

As Figure 7 shows, 1 μM KMUP-1 did not influence the SR Ca^2+^ release but significantly attenuated the Ca^2+^ entry via the SOCs in hypoxic PASMCs. Subsequently, KMUP-1 was co-incubated with the PKA inhibitor KT520 (1 μM), PKG inhibitor KT5823 (1 μM), and PKA activator PMA (1 μM). Those three agents did not affect the SR Ca^2+^ release but significantly restored Ca^2+^ entry via the SOCs. Those data further suggested that KMUP-1 retarded CCE through the SOCs involved in the PKA and PKG activation and PKC inhibition.

## 3. Discussion

The SOCE in PASMCs has gained substantial attention from research on hypoxic pulmonary vasoconstriction and pulmonary vascular remodeling. TRPC1 mainly contributes to SOCs, mediating SOCE induced by agonists or hypoxia [14,21,29]. In this study using hypoxic PASMCs, we first demonstrated that the xanthine derivative KMUP-1 inhibited TRPC1 expression and SOCs-mediated Ca^2+^ entry. This finding is consistent with previous findings [17,19,20] that KMUP-1 might be able to control cardiovascular diseases, particularly for the prevention or treatment of PAH.

A previous report showed that the levels of TRPC1 protein in PASMCs increased with the duration of hypoxia and reached a maximal level at 24 h [21]. We also found that the SR Ca^2+^ release and Ca^2+^ entry via the SOCs were significantly elevated under 24 h hypoxia incubations. Thus, we used the same conditions to observe changes in TRPC1 and Ca^2+^ entry via the SOCs by treatment with KMUP-1 in a hypoxia chamber for 24 h. Kunichika et al. [15] have demonstrated that TRPC1 in PASMC is an essential canonical TRP isoform that forms native SOC to regulate pulmonary vascular contractility. Our results also showed the relationship between TRPC1 and SOCE under hypoxic rat PASMCs ex vivo. Therefore, the data obtained from this study suggested that KMUP-1 hindered hypoxia-stimulating the expression of TRPC1 protein and associated Ca^2+^ entry via the SOCs, which could involve the PKA and PKG activation and the PKC inhibition. However, we cannot exclude that other factors and signaling pathways are involved in the effects of KMUP-1 in this rat model of hypoxic PASMCs.

SOCs are essential in controlling Ca^2+^ influx, arterial tone development, and smooth muscle cell growth in the pulmonary vasculature [10,11,12,13,30]. Few pulmonary VSM investigations have ventured beyond SOC modulation and explored the role of protein kinases [13,17,31]. Likewise, one of our previous reports demonstrated that PKA, PKG, and PKC regulate the activation of SOCs in pulmonary artery myocytes. PKA/PKG and PKC activation inhibited and stimulated SOCs, respectively [13]. As previously reported, KMUP-1 can modulate protein kinases A, G, and C [13]. Thus, we used PKA/PKG/PKC activators and inhibitors to study the effects of KMUP-1 on SR Ca^2+^ release and Ca^2+^ entry via the SOCs. In this study, we further confirmed that KMUP-1 attenuated TRPC1-dependent Ca^2+^ entry through the SOCs, which can be attributed to PKA/PKG activation and PKC inhibition. Notably, this finding can vary between different cell types and even within different subsets of smooth muscle cells. So, the actions of KMUP-1 on various smooth muscle cells under normoxic and hypoxic conditions need further investigation.

The cAMP/PKA signaling pathway is essential in several physiological processes, including vascular tone and cell cycle progression. Like PKA, cGMP/PKG relaxes vascular smooth muscle partly due to the inhibition of SOCs [13,30,31]. In contrast to PKA/PKG activation, PKC activation is strongly linked to increased SOC activity in vascular smooth muscle [13,31]. In physiological states, PKC activation would be more critical than PKA/PKG inhibition to induce vasoconstriction in PASMCs, stimulating the development of PAH. This study provides evidence that KMUP-1 is effective in preventing hypoxia-augmented TRPC1 expression and SOC-mediated Ca^2+^ entry in rat PASMCs (Figure 8).

Here, we want to address some of the limitations of this study. In general, the findings of this study were obtained from the SD rats and cannot be directly applicable to other species or human subjects. In addition, ex vivo PASMCs may not fully replicate the complex in vivo conditions within the PAs. The absence of interactions with neighboring cells, the lack of blood flow, and the absence of an extracellular matrix can influence cell behaviors. In other words, the behavior of PASMCs in culture may differ from their behavior in the intact PA due to alterations in the cellular microenvironment.

On the other hand, we have shown that KMUP-1 reduced the expression of the TRPC-1 protein in hypoxic conditions but did not measure its channel inhibitory activity on this current, which can be performed by patch-clamp electrophysiology in fresh PASMCs. In what way KMUP-1 modulates the SOCs-mediated Ca^2+^ influx upon SR Ca^2+^ depletion in hypoxic conditions, and the SOC channel inhibition by KMUP-1, still need to be considered. Moreover, TRPC1 activation of SOCE is not direct but needs the interaction of STIM1 and Orai1, which remains to be investigated by KMUP-1.

## 4. Materials and Methods

### 4.1. Animal Procedures and Tissue Preparation

The Kaohsiung Medical University Animal Care and Use Committee approved (IACUC No. 110140, 1 August 2022 to 31 July 2023) all procedures and protocols in compliance with the Guide for the Care and Use of Laboratory Animals published by the US National Institutes of Health. In brief, male Sprague-Dawley (SD) rats (13–15 weeks old) were euthanized by intraperitoneal injection (i.p.) of an overdose of pentobarbital sodium (130 mg/kg). The lungs and heart were cut by en bloc resection and put in a cold physiological salt solution (PSS) (in mM): 119 NaCl, 4.8 KCl, 1.7 KH_2_PO_4_, 20 NaHCO_3_, 10 Glucose, 1.2 CaCl_2_, and 1.2 MgSO_4_ (pH 7.4). Extralobar pulmonary arteries (PAs) were removed and free of the surrounding tissue.

### 4.2. Primary Culture of PASMCs

Extralobar PAs were dissected carefully and prepared for tissue culture. Ex vivo explant cultures were executed according to our previous reports. The vascular endothelium was removed using a sterile cotton swab with mild rubbing. Then, we carefully removed the tunica adventitia and the superficial part of the tunica media. The remaining pieces were cut into small pieces and then transferred to culture flasks for further cultivation. The ex vivo explants were incubated in Dulbecco’s Modified Eagle Medium (DMEM; Gibco Laboratories, Gaithersburg, MD, USA) with 10% fetal bovine serum and 1% penicillin-streptomycin amphotericin B (Biological Industries, Kibbutz Beit Haemek, Israel) to promote cell growth. The incubator was set to 37 °C with a humidified 5% CO_2_ environment. PASMCs began to proliferate from explants after culture for 7 days. The growth of PASMCs was arrested by substituting the media with FBS-free DMEM. Then, the cells were incubated in the absence and presence of test agents under normoxia (20% O_2_) or hypoxia (1% O_2_) for 24 h at 37 °C [21]. No more than five passages of primary cultures were used in the subsequent experiments. The immunofluorescent staining of *α*-actin was used to confirm the purity of PASMCs.

### 4.3. Western Blot Analysis

The expression of TRPC1 was determined by Western blotting. Briefly, PASMCs were lysed in M-PER (Mammalian Protein Extraction Reagent, 78501, Thermo Fisher Scientific, Waltham, MA, USA) comprising EDTA-free protease inhibitor cocktail and PhosSTOP phosphatase inhibitor (Roche Diagnostics, Mannheim, Germany). Next, the supernatants were collected after centrifugation at 15,000 rpm for 20 min at 4 °C. The Bradford assay was used to estimate the protein concentration. The protein was separated by sodium dodecyl sulfate–polyacrylamide gel electrophoresis (SDS-PAGE) and transferred to a polyvinylidene difluoride (PVDF) membrane. Subsequently, the membranes were blocked for 2 h at room temperature with 5% nonfat milk in Tris-buffered saline (20 mM Tris-HCl, 150 mM NaCl) with 0.1% Tween-20 (TBST), then incubated with primary antibodies against TRPC1 (1:200; #ACC-010, Alomone, Jerusalem, Israel) at 4 °C overnight. After TBST buffer washing three times, the membranes were incubated with horseradish peroxidase-conjugated secondary antibody. β-actin (1:5000, A5441; Sigma-Aldrich, St. Louis, MO, USA) was used as an internal control. Signals were detected with enhanced chemiluminescence (ECL) (Amersham Life Sciences Inc., Arlington Heights, IL, USA) HRP substrate reagent and quantized by densitometry with ImageJ software (version 1.53j; National Institutes of Health, Bethesda, MD, USA).

### 4.4. Measurement of Capacitative Ca^2+^ Entry

The intracellular Ca^2+^ concentrations ([Ca^2+^]_i_) were estimated using the fluorescent indicator Fura 2-AM (F0888, Sigma-Aldrich, St. Louis, MO, USA) and recorded by spectrofluorophotometer (Shimadzu, RF-5301PC, Kyoto, Japan). The preparation of PASMCs was excited at 340 and 380 nm to obtain the Fura 2 emission, and the resulting fluorescence emission at 510 nm was monitored. The ratio of emissions at 510 nm (F340/F380) was recorded every 2 sec based on the equation [Ca^2+^]_i_ = Kd × (Sf2/Sb2) × (R − R_min_)/(R_max_ − R). Depletion of Ca^2+^ from the ER/SR triggered the SOCs opening in the plasma membrane, so-called capacitative Ca^2+^ entry (CCE) [15]. For measuring CCE, PASMCs were loaded with 5 µM Fura 2-AM for 30 min at 37 °C in the dark and then washed with PSS twice to eliminate the extracellular Fura 2. Then, the PASMCs were put in Ca^2+^-free PSS and supplied with 1 µM sarco/endoplasmic reticulum Ca^2+^-ATPase (SERCA) inhibitor thapsigargin (TG) to deplete intracellular Ca^2+^ stores and 5 µM nifedipine to block the voltage-operated Ca^2+^ channels (VOCC), followed by extracellular Ca^2+^ (2.5 mM) to initiate CCE.

### 4.5. Chemicals

Buffer reagents, 8-Br-cAMP, 8-Br-cGMP, chelerythrine, Fura 2-AM, KT5823, KT5720, phorbol 12-myristate 13-acetate (PMA), and thapsigargin were purchased from Sigma-Aldrich Chemical Co. (St. Louis, MO, USA). Unless specified otherwise, all drugs and reagents were dissolved in distilled water. Fura 2-AM, KT5720, KT5823, PMA, and thapsigargin were dissolved in DMSO at 10 mM; KMUP-1 was dissolved in 10% absolute alcohol, 10% propylene glycol, and 2% 1N HCl at 10 mM. Serial dilutions were performed in phosphate buffer solution, with the final solvent concentration < 0.01%.

### 4.6. Statistical Analysis

All data are shown as the mean ± SE. To identify statistical variances, independent and paired Student’s *t*-tests were conducted on unpaired and paired samples. The one-way analysis of variance (ANOVA) was employed when multiple treated groups were compared to a control group. When the ANOVA manifested a statistical difference, a Tukey–Kramer pairwise comparison was used for post hoc analysis in cases where applicable. A probability value (*p*-value) less than 0.05 was considered significant.

## 5. Conclusions

KMUP-1 attenuated TRPC1 protein and SOC-dependent Ca^2+^ entry in this hypoxic PASMCs model, which could be attributed to its PKA and PKG activation and PKC inhibition. In addition to the benefits of monocrotaline-induced PAH by KMUP-1 [25,26], it also could cause PA smooth muscle relaxation in the model of hypoxia-induced PAH. Finally, we suggest that KMUP-1 can be a potential therapeutic candidate for targeting hypoxia-induced PAH.

## Figures and Tables

**Figure 1 pharmaceuticals-17-00440-f001:**
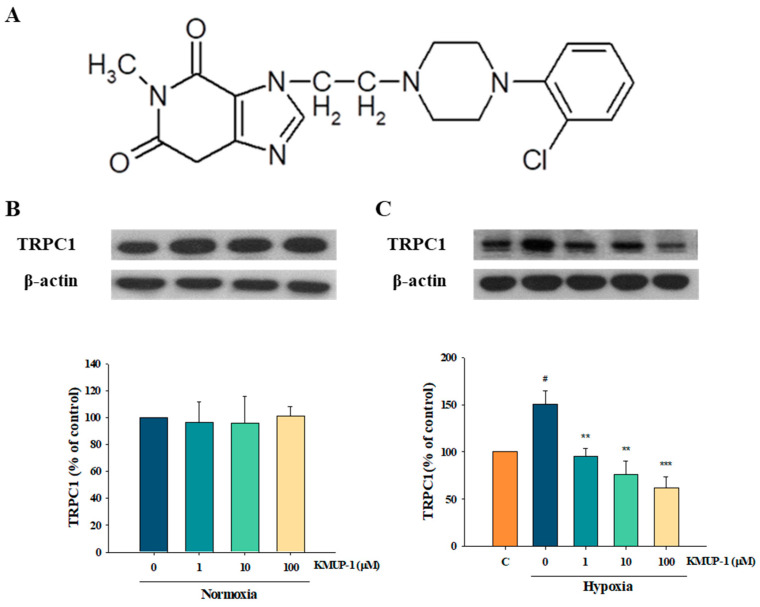
Effects of KMUP-1 in normoxic and hypoxic pulmonary arterial smooth muscle cells (PASMCs). (**A**) Structure of KMUP-1. (**B**) TRPC1 protein showed no significant differences in KMUP-1 (1, 10, 100 μM)-treated PASMCs under normoxic conditions. (**C**) Various concentrations of KMUP-1 (1, 10, 100 μM) inhibited hypoxia-induced TRPC1 protein overexpression. The quantitation of this protein is shown in the lower panel. Results were presented as the mean ± SE, n = 7. ^#^
*p* < 0.05 compared with control (normoxia), r = 0.66; ** *p* < 0.01, *** *p* <0.001 compared with hypoxia, r = 0.81. C: control represents normoxia. r: correlation coefficient.

**Figure 2 pharmaceuticals-17-00440-f002:**
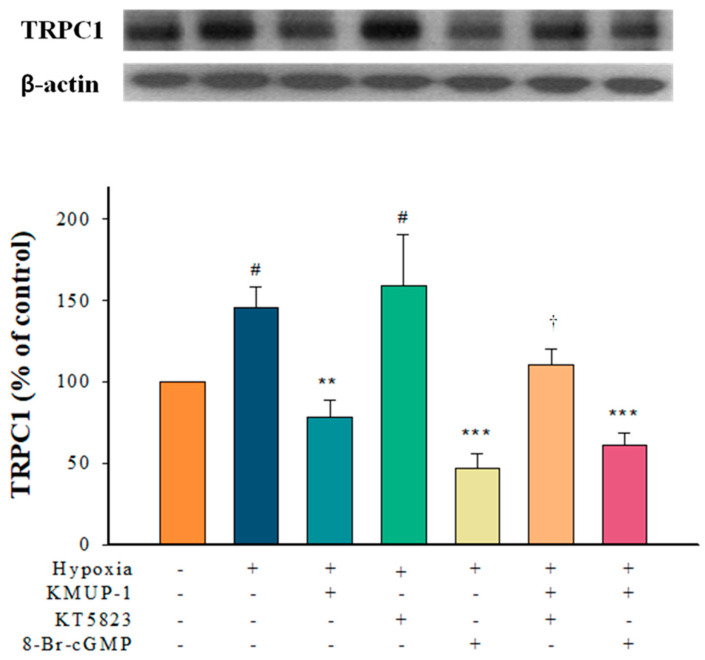
KMUP-1 inhibited hypoxia-enhanced TRPC1 expression via the cGMP/PKG pathway. PASMCs pretreated with KMUP-1 (1 μM), KT5823 (1 μM), 8-Br-cGMP (100 μM), KT5823+KMUP-1, and 8-Br-cGMP+KMUP-1 under hypoxic states. The quantitation of these proteins is shown in the lower panel. Results are presented as the mean ± SE, n = 6. ^#^
*p* < 0.05 compared with normoxia, r = 0.47; ** *p* < 0.01, *** *p* < 0.001 compared with hypoxia, r = 0.72; ^†^ *p* < 0.05 compared with KMUP-1 group, r = 0.56. 8-Br-cGMP: membrane-permeable analog of cGMP. r: correlation coefficient.

**Figure 3 pharmaceuticals-17-00440-f003:**
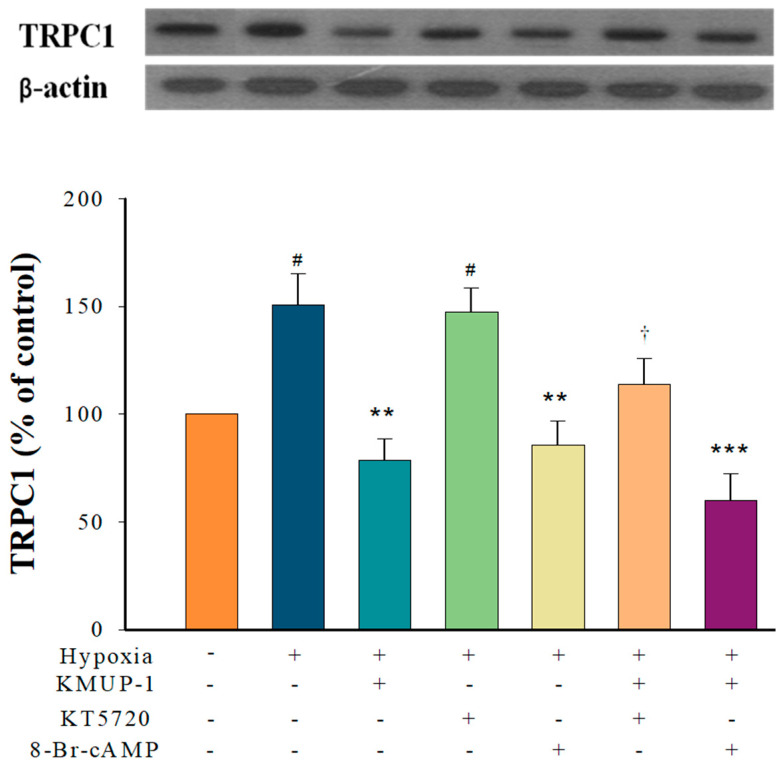
KMUP-1 inhibited hypoxia-enhanced TRPC1 expression via the cAMP/PKA pathway. PASMCs pretreated with KMUP-1 (1 μM), KT5720 (1 μM), 8-Br-cAMP (100 μM), KT5720+KMUP-1, and 8-Br-cAMP+KMUP-1 under hypoxic states. The quantitation of these proteins is shown in the lower panel. Results are presented as the mean ± SE, n = 6. ^#^
*p* < 0.05 compared with normoxia, r = 0.59; ** *p* < 0.01, *** *p* < 0.001 compared with hypoxia, r = 0.68; ^†^
*p* < 0.05 compared with KMUP-1 group, r = 0.61. 8-Br-cAMP: membrane-permeable analog of cAMP. r: correlation coefficient.

**Figure 4 pharmaceuticals-17-00440-f004:**
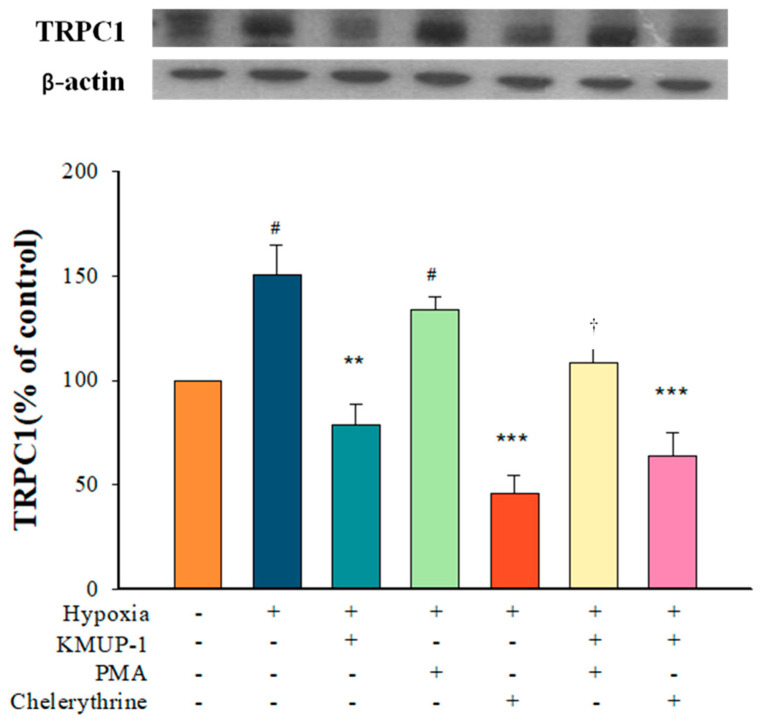
KMUP-1 inhibited hypoxia-enhanced TRPC1 expression via PKC pathway. PASMCs pretreated with KMUP-1 (1 μM), PMA (1 μM), chelerythrine (1 μM), PMA+KMUP-1, and chelerythrine+KMUP-1 under hypoxic states. The quantitation of these proteins is shown in the lower panel. Results are presented as the mean ± SE, n = 6. ^#^
*p* < 0.05 compared with normoxia, r = 0.48; ** *p* < 0.01, *** *p* < 0.001 compared with hypoxia, r = 0.70; ^†^ *p* < 0.05 compared with the PMA group, r = 0.52. PMA: phorbol 12-myristate 13-acetate. r: correlation coefficient.

**Figure 5 pharmaceuticals-17-00440-f005:**
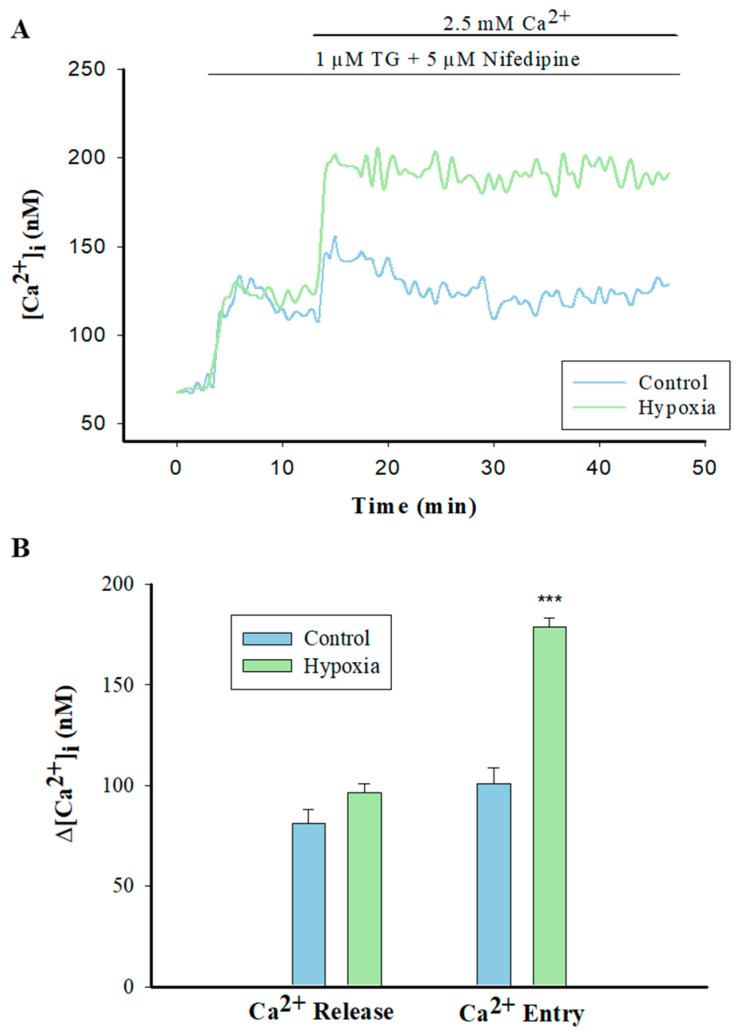
Effects of exposure to hypoxia on capacitative calcium entry in PASMCs. (**A**) Change in [Ca^2+^]_i_ in pulmonary arterial smooth muscle cells (PASMCs) from control or hypoxia and subjected to the restoration of extracellular Ca^2+^ following store depletion with TG (1 μM). (**B**) Bar graph illustrates mean ± SE change in [Ca^2+^]_i_ (∆[Ca^2+^]_i_) in response to TG and Ca^2+^ restoration. All experiments were performed in the presence of nifedipine, n = 12 for control and n = 13 for hypoxia. *** *p* < 0.001 compared with control (normoxia). TG: thapsigargin.

**Figure 6 pharmaceuticals-17-00440-f006:**
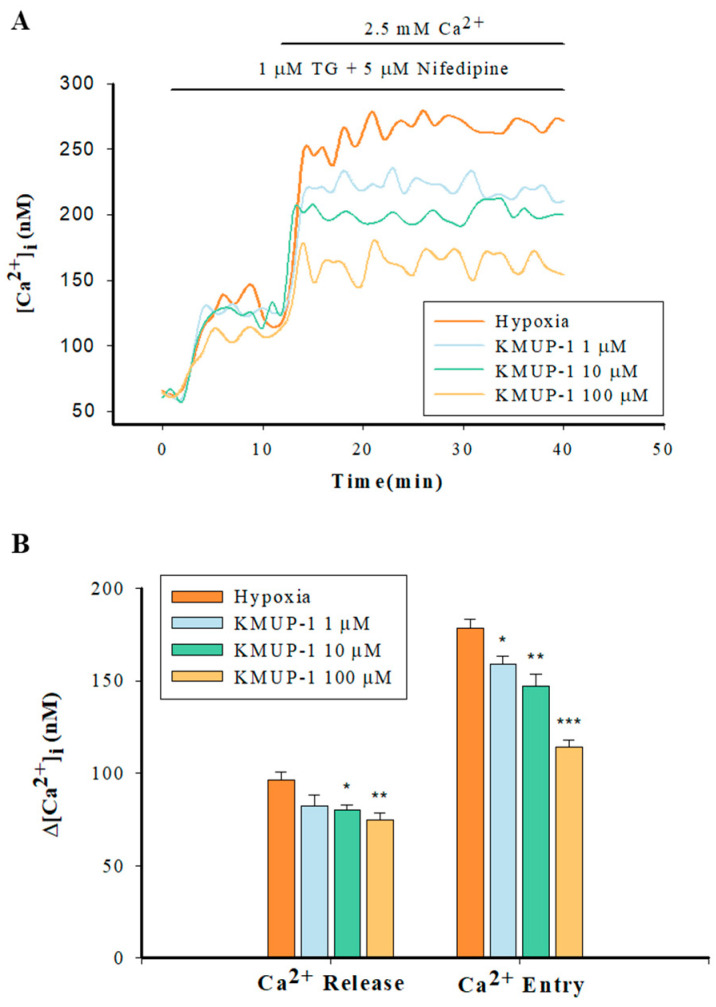
Effects of KMUP-1 on capacitative calcium entry in PASMCs. (**A**) Change in [Ca^2+^]i in pulmonary arterial smooth muscle cells from hypoxic or pretreated KMUP-1 groups and subjected to the restoration of extracellular Ca^2+^ following store depletion with TG (1 μM). (**B**) Bar graph illustrates mean ± SEM change in [Ca^2+^]_i_ (∆[Ca^2+^]_i_) in response to TG and Ca^2+^ restoration. All experiments were performed in the presence of nifedipine, n = 7–13 of independent experiments. * *p* < 0.05, ** *p* < 0.01, *** *p* < 0.001 compared with hypoxic group. TG: thapsigargin.

**Figure 7 pharmaceuticals-17-00440-f007:**
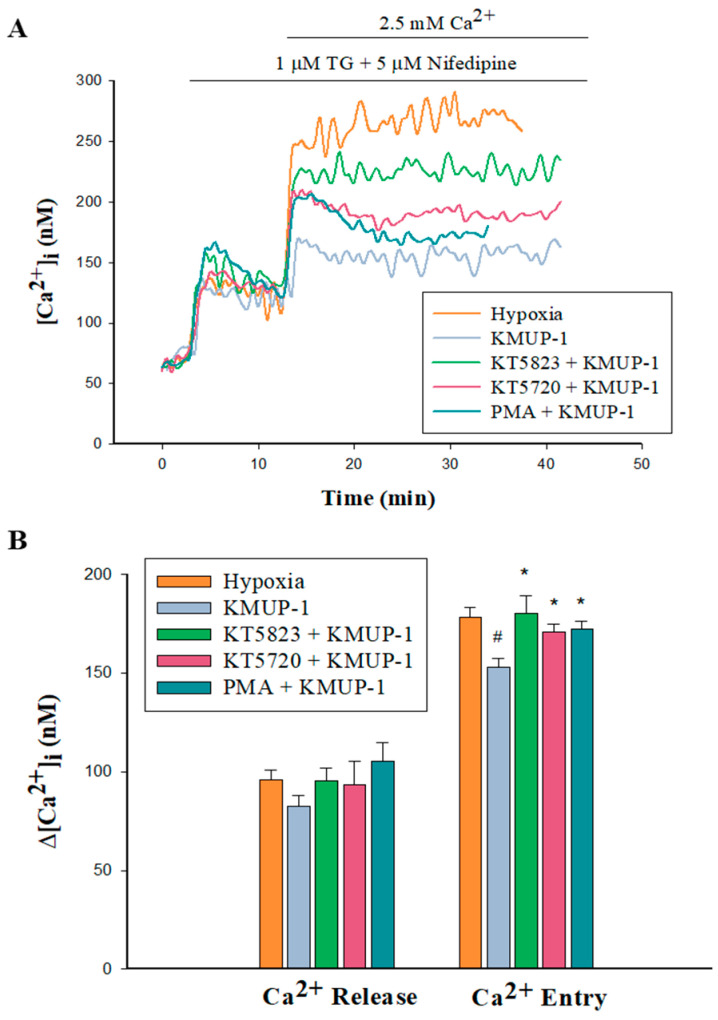
Effects of KT5823, KT5720, and PMA on KMUP-1-inhibited capacitative calcium entry in hypoxic PASMCs. (**A**): Change in [Ca^2+^]_i_ in pulmonary arterial smooth muscle cells (PASMCs) from pretreated KMUP-1 (1 μM), KT5823 (1 μM) with KMUP-1, KT5720 (1 μM) with KMUP-1, PMA (1 μM) with KMUP-1 and then subjected to the restoration of extracellular Ca^2+^ following store depletion with TG (1 μM). (**B**): Bar graph illustrates mean ± SE change in [Ca^2+^]_i_ (∆[Ca^2+^]_i_) in response to TG and Ca^2+^ restoration. All experiments were performed in the presence of nifedipine, n = 6 of independent experiments. ^#^
*p* < 0.05 compared with hypoxic group; * *p* < 0.05 compared with KMUP-1 group in hypoxic PASMCs. PMA: phorbol 12-myristate 13-acetate; TG: thapsigargin.

**Figure 8 pharmaceuticals-17-00440-f008:**
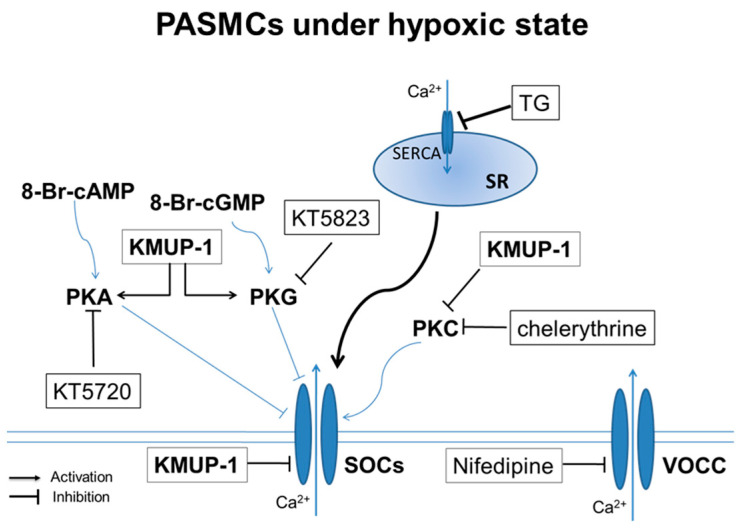
Diagram summarizing the actions of KMUP-1 on the store-operated calcium channel (SOCs) in hypoxic PASMCs. Data suggest that KMUP-1 inhibits TRPC1 encodes for SOCs, activates the PKA/PKG pathway, and inhibits the PKC pathway. SOCs: store-operated calcium channels; SR: sarcoplasmic reticulum; SERCA: sarco/endoplasmic reticulum Ca^2+^-ATPase; TG: thapsigargin; PMA: phorbol 12-myristate 13-acetate; 8-Br-cAMP: membrane-permeable analog of cAMP; 8-Br-cGMP: membrane-permeable analog of cGMP; VOCC: voltage-operated calcium channels.

## Data Availability

The data supporting this study’s findings are available from the corresponding author upon request.

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
