# Peer review of "The Xanthine Derivative KMUP-1 Inhibits Hypoxia-Induced TRPC1 Expression and Store-Operated Ca2+ Entry in Pulmonary Arterial Smooth Muscle Cells"

_pharmaceuticals, 2024, doi:10.3390/ph17040440_

Round 1

Reviewer 1 Report

Comments and Suggestions for Authors

Title: The Xanthine Derivative KMUP-1 Inhibits Hypoxia-Induced 2 TRPC1 Expression and Store-Operated Ca2+ Entry in Pulmonary 3 Arterial Smooth Muscle Cells.

Recommendation for publication after addressing the following comments.

1. The abstract could be more succinctly structured to enhance readability. It should maintain a balance between providing essential information and avoiding unnecessary detail.

2. Simplify or explain technical terms and abbreviations where appropriate to ensure accessibility to a wider audience.

3. Consider including key findings or quantitative data to provide a clearer picture of the results obtained.

4. Expand on the potential clinical or therapeutic implications of the findings to enhance the relevance and impact of the study

5. Review the abstract for grammatical errors and refine sentence structure to improve clarity and flow.

6. Ensure that the conclusion accurately reflects the study's main findings and their significance.

7. While the introduction covers various aspects of PAH and the role of SOCs and TRPC1, it could be more concise. Focus on key points and avoid repetition to streamline the text.

8. Use transition phrases to guide the reader through different sections of the introduction.

9. Ensure that technical terms are explained or defined for readers who may not be familiar with the terminology.

10. Highlight the significance of the study earlier in the introduction to engage readers and underscore the importance of the research.

11. Strengthen the connection between previous research findings and the current study. Provide more context on how the findings of this study build upon existing knowledge and address gaps in the literature.

12. Some sentences are lengthy and could be simplified for clarity in the discussion.

13. While the discussion outlines the effects of KMUP-1 on TRPC1 expression and SOCE modulation, it lacks detailed explanations of the underlying mechanisms.

14. Strengthen the discussion by integrating findings from previous studies that support or contrast with the current findings.

15. Acknowledge any limitations of the study, such as experimental constraints or potential confounding factors, and suggest avenues for future research to address these limitations.

Comments on the Quality of English Language

Moderate editing of English language required

Author Response

The authors would like to thank the Editors and the reviewers for their positive support of our work. We appreciate the comments on our study, which have enabled us to improve this manuscript. We are happy to follow the suggestions and have made the necessary amendments for the final revision. We hope our revised manuscript is now acceptable for publication in “Pharmaceuticals.”

We appreciate your efforts in reviewing our paper and providing insightful suggestions. Our point-to-point responses are listed below.

  1. Reply: The Abstract has been carefully revised to improve its readability. So far, it should be more accessible to comprehend.

  1. Reply: We have provided and explained the professional terms to ensure readers’ accessibility and understanding.

  1. Reply: Regarding your concerns, we have provided a Graphical Abstract to include our key findings. Thanks for your suggestion.

  1. Reply: We have expanded the potential therapeutic implications of the findings. Please take a look, thanks.

  1. Reply: The Abstract for grammatical errors and typos has been fixed.

  1. Reply: The conclusion has been revised to reflect the main findings, and the text is highlighted. Please take a quick look; thanks.

  1. Reply: We have almost rewritten the section of the Introduction. The role of SOCs and TRPC1 has been revised to clarify it. Since the editorial office wants me to increase the word count (~4000), I have added some descriptions to the section. I’m sorry about that. All I have added has been highlighted in red.

  1. Reply: As per your suggestion, the Introduction section has used some transition phrases to make the reader easy to follow.

  1. Reply: We have explained and defined the professional terms to ensure readers’ accessibility and understanding.

  1. Reply: Thanks for your great suggestions. We have highlighted the significance of the earlier study in the Introduction section to attract the readers and emphasize the importance of this work.

  1. Reply: We have strengthened the connection of our work to previous research findings in the Introduction and Discussion sections. The new descriptions added have been highlighted in red.

  1. Reply: We have clarified some lengthy sentences in the Discussion section.

  1. Reply: In Discussion: The effect of KMUP-1 on TRPC1 expression and SOCE modulation, we have addressed some essential information to explain the underlying mechanism of KMUP-1. Please see the sentences highlighted inRepy red.

  1. Reply: We have strengthened the Discussion section to discuss the previous findings and our article. Thank you so much for your great suggestions.

  1. We have added some limitations of this study in the last paragraph of the Discussion sections. The text has been highlighted in red.

Reviewer 2 Report

Comments and Suggestions for Authors

Summary of the Work

The general objective of this work is to study the pathogenesis and pathophysiology of pulmonary arterial hypertension (PAH), characterized by progressive pulmonary arterial resistance and increased pressure. This study, in particular, aims to search for and develop new agents that can target PAH. More in particular, the authors investigate how the K+ channel activator KMUP-1 modulates protein kinase to protect against hypoxic pulmonary artery smooth muscle cells (PASMCs) that mimic PAH.

Main Results Obtained

The authors

i) showed that using hypoxic PASMCs, the xanthine derivative KMUP-1 modulates the transient receptor potential channel 1 (TRPC1) expression and the tore-operated calcium channels (SOCs)-mediated Ca2+ entry;

ii) suggested that KMUP-1 hindered hypoxia-stimulated TRPC1 expression through protein kinase A (PKA) and the protein kinase G (PKG) activation and the protein kinase C (PKC) inhibition;

iii) found that KMUP-1 attenuates Ca2+ entry through the SOCs and that this can be attributed to PKA/PKG activation and PKC inhibition.

The above results suggest that KMUP-1 is a potential therapeutic candidate for targeting hypoxia-induced PAH.

General Considerations

- English should be double-checked; several typos were found.

- All acronyms must be specified when they appear for the first time in the manuscript, even if they are well-known in the literature. The abstract contains many acronyms not specified (e.g., KMUP-1 = a xanthine derivative with phosphodiesterase inhibitor activity, Ca2+ = The calcium ion, Fura-2/AM, cGMP, etc.)

- The statistical analysis of the obtained results is missing (see the suggestions below).

- While studies investigating the relationship between hypoxia-induced TRPC1 expression and store-operated calcium entry in pulmonary arterial smooth muscle cells (PASMCs) provide valuable insights, it's also true that there are limitations and potential objections associated with such results. These limitations have not been exhaustively discussed in this study.

- Anyhow, the manuscript is interesting and contains a lot of information.

The following suggestions are intended to help fill some gaps

Suggestions

1) The method used to measure Ca2+ release and entry should be critically evaluated. The authors are asked to clarify how they managed issues linked to fluorescent dyes and genetically encoded indicators. Furthermore, it is not clear to me how they have managed technical issues related to sensitivity, specificity, and potential interference with cellular functions. Please, clarify.

2) Although it is possible to observe overexpression of TRPC1, directly linking it to functional outcomes, such as changes in Ca2+ release and entry, interpretation of the results still requires careful attention. In other words, how can the authors ensure that other factors and signalling pathways are not also involved in the process?

3) As known, while p-values are commonly used to determine statistical significance, it's important to recognize that relying solely on p-values can be limiting. In particular, a low p-value alone doesn't provide information about the practical significance or effect size. Please provide the values of the correlation coefficients related to the KMUP-1 effects on normoxic and hypoxic pulmonary arterial smooth muscle cells (PASMCs), and to KMUP-1 inhibited hypoxia-enhanced TRPC1 expression (via the cGMP/PKG pathway, the cAMP/PKA pathway, and the PKC pathway).

4) We also know that alongside point estimates, confidence intervals provide a range of values within which the true parameter is likely to fall. Please specify, more precisely, the confidence intervals for the KMUP-1 experiments shown in Figures 1., 2., 3., and 4.

5) The responses to hypoxia and the expression of TRPC1 can vary between different cell types and even within different subsets of smooth muscle cells. We may object that extrapolating findings from PASMCs to other cell types or vascular beds may lead to incorrect results. The authors are invited to dispel this possible objection.

6) To elucidate TRPC1 overexpression and observe the Ca2+ release and entry, the authors cultured primary pulmonary artery smooth muscle cells (PASMCs) from Sprague-Dawley rats in a modular incubator chamber under 1% O2/5% CO2 for 24 h. However, there are some limitations associated with this method that need to be duly highlighted. For instance,

6a) Results obtained from cell culture may not fully replicate the complex in vivo conditions within the pulmonary arteries. The absence of interactions with neighboring cells, the lack of blood flow, and the absence of the extracellular matrix can influence cell behavior.

6b) The process of isolating and culturing primary cells may introduce variability. The behavior of PASMCs in culture may differ from their behavior in the intact pulmonary artery due to changes in the cellular microenvironment.

The authors are asked to reply to the limitations mentioned in the above points 3a) and 3b).

7) Another major limitation of the method proposed by the authors arises from the fact that their results refer to the specificity of the rat model. How can the authors claim that the results of the Sprague-Dawley rat studies can be directly applicable to humans or other species? Indeed, in principle, there could be species-specific differences in the response to hypoxia and TRPC1 overexpression.

Conclusions

The work is interesting, topical, and contains a lot of information; I enjoyed reading it. However, it shows some vulnerable points and gaps that need to be filled, such as the lack of exhaustive statistical analysis and a deep discussion on the limitations related to the approach followed by the authors (e.g., the limitations related to the rat model specificity, etc.). The authors are highly advised to take into account the suggestions expressed above.

Comments on the Quality of English Language

English should be double-checked; several typos were found.

Author Response

The authors would like to thank the Editors and the reviewers for their positive support of our work. We appreciate the comments on our study, which have enabled us to improve this manuscript. We are happy to follow the suggestions and have made the necessary amendments for the final revision. We hope our revised manuscript is now acceptable for publication in “Pharmaceuticals.”

We appreciate your effort, thorough review, and helpful suggestions. As follows, we respond to your concerns by point-to-point.

General considerations

  • Reply: Several typos have been fixed, thanks.
  • Reply: We have specified all acronyms while they appear for the first time.
  • Reply: This revised manuscript will provide the values of the correlation coefficients (r).
  • Reply: We have addressed the limitations in this revised manuscript.
  • Reply: Thanks for your kind consideration and help filling this article’s gaps.

Suggestions

  • Reply: Thank you for your concerns. We used cell suspension (fixed cell numbers) methods to measure the intracellular calcium concentrations. We have done such kinds of experiments ordinarily. We don’t have the problem of sensitivity, specificity, or interference with cellular function.

  • Reply: Thanks for your comments. However, we cannot exclude that other factors and signaling pathways are involved in the effects of KMUP-1 in this rat model of hypoxic PASMCs. We have added this insightful comment to the Discussion part of this article.

  • Reply: Sure, we have provided the values of the correlation coefficients (r) in the figure legend of this revised manuscript. Please take a look, and thanks.

  • Reply: Per your concerns, the confidence intervals can be calculated at different significance levels. We use α to denote the significance level and perform a hypothesis test with a 100 (1−α)% confidence interval. Based on such concepts, the packaged software (SigmaStat or GraphPad InStat) runs the significance of the p-value.

  • Reply: Thanks for your invaluable suggestions. We have added the concepts to our Discussion section. The descriptions are as follows. Notably, this finding can vary between different cell types and even within different subsets of smooth muscle cells. So, the actions of KMUP-1 on various smooth muscle cells under normoxic and hypoxic conditions need further investigation.

  • Reply 6a) and 6b): We agree with your opinions. Data obtained from cell culture may not fully replicate the complex in vivo conditions within the pulmonary arteries. We’ll address this matter in the limitations of this study.

  • Reply: Of course, the study of animal models cannot reflect other species and human subjects. We’ll carefully revise this article and make a note in the limitation section. 

Reviewer 3 Report

Comments and Suggestions for Authors

The article covers an interesting and current topic. Nevertheless, in my opinion, some parts need to be improved, I have some comments:

1- We also used Fura-2/AM to measure the stored calcium release from the sarco- 28 plasmic reticulum (SR) and calcium entry through SOCs in hypoxic PASMCs. Under hypoxic con- 29 ditions, store-operated calcium entry (SOCE) activity was enhanced in PASMCs, and KMUP-1 could 30 refill the stores of SR and attenuate SOCE activity. In summary, KMUP-1 inhibited TRPC1 protein 31 and reduced SOCE, possibly through the SOCs in hypoxic PASMCs. Abstract might be beneficial to include a sentence that briefly summarizes the key findings of the study. This can provide readers with a quick overview of the research. 

2- 1. Introduction 36 Pulmonary arterial hypertension (PAH), characterized by progressive pulmonary ar- 37 terial resistance and pressure elevation, has a poor prognosis. The pathogenesis and path- 38 ophysiology of PAH remain elusive. They include a rare dyspnea-fatigue syndrome due 39 to a progressive increase in pulmonary vascular resistance (PVR) and eventually right 40 ventricular (RV) failure [1]. The vascular lesions in PAH are characterized by vasocon- 41 traction, smooth muscle and endothelial cell proliferation, and thrombus formation [2]. Although the Authors described in detail the findings from the included references, there are several relevant works/reviews, which should be added and discussed by the Authors:

a- Recent Advances in the Treatment of Pulmonary Arterial Hypertension. Pharmaceuticals 202215, 1277. https://doi.org/10.3390/ph15101277

b-An Overview of Different Techniques for Improving the Treatment of Pulmonary Hypertension Secondary in Systemic Sclerosis Patients. Diagnostics (Basel). 2022 Mar 1;12(3):616. doi: 10.3390/diagnostics12030616.

3- This study inves- 86 tigates how the K+ channel activator KMUP-1 modulates protein kinase to protect against 87 hypoxic PASMCs that mimic PAH. Please improve the description of this part and underline the novelty of the study.

4- 2. Results. The paragraph is quite rumbling and difficult to read. Please, clarify and underline the most important statistically significant data to better support the conclusions.

5- 3. Discussion 205 SOCE in PASMCs have gained substantial attention from research on hypoxic pul- 206 monary vasoconstriction and pulmonary vascular remodeling. TRPC1 mainly contributes 207 to SOCs, mediating SOCE induced by agonists or hypoxia [13,31,32]. In this study using 208 hypoxic PASMCs, we have first demonstrated that the xanthine derivative KMUP-1 mod- 209 ulated TRPC1 expression and SOCs-mediated Ca2+ entry. The discussion section needs to be improved.  It could be interesting to record the aim of the study. It is necessary to compare the data  with previous published literature. 

6- 4.6. Statistical analysis 306 All data are expressed as the mean ± SE. Data were compared using the unpaired 307 Student’s t-test. One-way analysis of variance was used to analyze differences among 308 multiple comparisons. When appropriate, a Tukey-Kramer pairwise comparison was 309 used for post hoc analysis. Differences were considered to be statistically significant when 310 P < 0.05. please improve the description of the tests used to evaluate the data. 

7- 5. Conclusions 313 KMUP-1 attenuated TRPC1 protein and inhibited SOC-dependent Ca2+ entry in hy- 314 poxic PASMCs. Since this would result in smooth muscle relaxation in pulmonary arter- 315 ies, we suggest that KMUP-1 is a potential therapeutic candidate for targeting hypoxia- 316 induced PAH. Please improve the conclusions and underline the novelty of the study and clarify the possible clinical implications of the observations. 

Comments on the Quality of English Language

Minor changes of English language are required

Author Response

The authors would like to thank the Editors and the reviewers for their positive support of our work. We appreciate the comments on our study, which have enabled us to improve this manuscript. We are happy to follow the suggestions and have made the necessary amendments for the final revision. We hope our revised manuscript is now acceptable for publication in “Pharmaceuticals.”

We appreciate your extensive efforts in reviewing our article and providing invaluable suggestions. As follows, our responses are point to point, as you are concerned.

  1. Abstract: Thanks for your suggestions. We agreed that the Abstract should include a sentence briefly summarizing the study’s key findings. The Abstract has been carefully revised to improve its readability in this revised manuscript. So far, it should be more accessible to comprehend. The revised text has been highlighted in red. Please take a look at this section; thanks again.

  1. Introduction: Thanks for your invaluable suggestions and provides me with some relevant articles on PAH; I appreciate it. The article you suggested has been cited in this revised manuscript. We have added some descriptions in the Introduction and Discussion sections. All added texts have been highlighted in red.

  1. Thanks for your suggestions. We have addressed many descriptions regarding the K+ channel activator KMUP-1 and how to modulate protein kinase to protect against hypoxic PASMCs. We also added the limitations of this study in this revised manuscript. We hope that this revised manuscript has satisfied your concerns.

  1. Results: We are sorry for the carelessness of our previous manuscript version. To meet the format of this journal, we copied and pasted the wrong information into the section. No matter it is hard to read. We have fixed this problem in this revised manuscript. We also carefully revised this part to make it easy to read and comprehend. Thanks so much for your efforts in reading this article.

  1. Discussion: Thank you very much for your insightful suggestion. We have an effort to work on this part. Some relevant reports have been cited to discuss and support our data. You’ll find this part has been clarified, and the limitation of this study is also included. Please see the text in red. We hope that has satisfied your concerns.

  1. Statistical analysis: Thanks for your comments. We have rewritten the section on statistical analysis and added some descriptions to clarify it. Please take a look at this part.

  1. Conclusions: Thank you for your suggestions. We have revised this part and made it clear and precise. Our conclusions are as follows. KMUP-1 attenuated TRPC1 protein and SOC-dependent Ca2+ entry in this hypoxic PASMCs model, which could be attributed to its PKA and PKG activation and PKC inhibition. In addition to the benefits of monocrotaline-induced PAH by KMUP-1 [25,26], it also could have PA smooth muscle relaxations in the model of hypoxia-induced PAH. Finally, we suggest that KMUP-1 can be a potential therapeutic candidate for targeting hypoxia-induced PAH. Thanks again.

Round 2

Reviewer 1 Report

Comments and Suggestions for Authors

Aceepted

Reviewer 2 Report

Comments and Suggestions for Authors

The authors have taken into account all the suggestions made in my previous report. In my opinion, this revised version deserves to be published.

Reviewer 3 Report

Comments and Suggestions for Authors I have carefully evaluated the revised manuscript. The authors adequately answered my questions. In my opinion this improved the manuscript. I have no further comments.

Comments on the Quality of English Language

 Minor changes of English language are required